# Teaching Systems-Thinking Concepts with Hypothetical Case Scenarios: An Exploration in Food-Systems Science Education

**DOI:** 10.3390/foods12142663

**Published:** 2023-07-11

**Authors:** Catherine E. Sanders, Allison R. Byrd, Kristin E. Gibson, Aaron Golson, Kevan W. Lamm, Alexa J. Lamm

**Affiliations:** 1Department of Agricultural and Human Sciences, College of Agriculture and Life Sciences, North Carolina State University, Raleigh, NC 27695, USA; 2Department of Agricultural Leadership, Education, and Communication, College of Agricultural and Environmental Sciences, University of Georgia, Athens, GA 30602, USA; afortner@uga.edu (A.R.B.); kristin.gibson@uga.edu (K.E.G.); aaron.golson@uga.edu (A.G.); kl@uga.edu (K.W.L.);

**Keywords:** sustainable seafood, higher education pedagogy, sustainable development goals, food systems education, curriculum development

## Abstract

Background: The ability to solve complex challenges facing the food system is an increasingly important skill for undergraduate students entering into the global workforce. However, the curriculum tools to enhance systems thinking in the undergraduate agricultural and natural resource classroom are limited, specifically related to food systems education. Methods: The current study explored the use of hypothetical case scenarios in a teaching curriculum related to the seafood industry, in order to determine the use of these tools as a mechanism for increasing undergraduate students’ systems-thinking capacity. The mixed-method study used a survey and focus groups. Results indicated that participants’ understanding of, and capacity for, using systems thinking to reason through complex scenarios increased during the mixed-method educational intervention. Participants stated the experience helped them learn more about their own patterns of thinking, created transformational learning moments through dissonance, helped them learn about the unintended consequences of decision-making, and increased their understanding of system complexity. Conclusions: The implications of using hypothetical case scenarios in the food system education classroom are explored.

## 1. Introduction

Topics within agricultural and food systems education are complex, compounded by the information-dense environment in which students and the public find themselves daily [1]. Additionally, the science–society relationship has become increasingly complex and interconnected [2], leading to more complex and controversial engagements with socio-scientific issues (SSIs). SSIs are “complex, ill-defined, critical societal issues that have a basis in science” [3] (p. 1339), such as climate change and genetically modified organisms. One component of learning about SSIs, including those associated with the food system, is developing the capacity for systems thinking, specifically related to the social dimensions of SSIs [3].

### 1.1. Systems Thinking as a Method for Sustainable Food Systems Education

Systems thinking, related to complex systems learning, is an increasingly popular learning outcome within science education [4]. Integrating systems thinking into scientific learning processes can help students use relevant theories, and develop problem-solving skills, focused on current global issues, specifically related to the interdisciplinary nature of scientific practice and the food system, based on the three pillars of sustainability (social, environmental, and economic) [5,6]. Systems thinking can help students reason about the scientific, socioeconomic, cultural, and political dimensions of complex challenges, harnessing an analytical focus toward resolving the underlying causes of, rather than surface-level adaptations to, issues [3,7]. However, many students do not have the requisite skills necessary to reason through issues systemically, leading them to misidentify causes, and to fail to fully appreciate the complexity of issues [3,8]. One way to address this deficit in systems-thinking skill development is to provide students with the tools to facilitate and support systems thinking in the classroom [9].

One of the primary challenges, however, for instructors implementing SSI-related content, such as systems thinking, in the classroom, is a lack of pedagogical tools and professional development competencies to facilitate systems-orientated learning [3,10]. While applying course concepts specifically related to SSIs to real-world situations can promote diverse thinking, leading to more innovative ideas for implementing solutions to complex challenges facing the food system [5,11], few strategies exist for instructors to implement this in the classroom. Systems thinking can be used to scaffold educational concepts, in science and food systems education, to create “a positive, open, and free-thinking learning environment allow[ing] students to create and encourag[ing] them to take risks as they learn” [5] (p. 2853). Generally, the generation of solutions for SSIs requires the evaluation of trade-offs, and a comparison of the benefits and disadvantages among various potential solutions [12,13]. Crafting opportunities for evaluating trade-offs within the classroom can help connect course content with real-world situations, to engage students in the related decision-making processes [5,13]. One method of creating educational opportunities for such decision making, enhancing both cognitive and affective connections in the learning process, is the use of hypothetical case scenarios (HCSs) [14,15]. HCSs are a form of learner-directed and solution-choice activities, known as choose-your-own-adventure activities, which help enhance student engagement in the learning process, through the gamification of learning [14,16,17]. HCSs also offer opportunities for students to engage in risk-taking within learning, helping to foster creativity within the systems-based learning process [5].

Most research related to SSIs and systems learning in the classroom has focused on the acquisition of content knowledge, and the alignment with learning outcomes, such as understanding scientific concepts, models, and methods of reasoning [3,18]. Fewer studies have focused on student learning processes, metacognition, and experiences with learning tools using SSI topics and systems learning environments [3]. Additionally, there is a dearth of teaching tools to facilitate systems learning for students within the food system context, specifically across the content areas represented in the United Nations Sustainable Development Goals (SDGs). The current study presents a mixed-method perspective of students’ experiences with a HCS, systems-based learning tool, to provide insight into the use of systems-based pedagogy in the higher education classroom, with a particular focus on food systems.

### 1.2. Study and Educational Context: Sustainable Marine Eco-Food Systems

The topical context area for the HCSs used in the current study was focused on the environmental and social impacts that humans have when interacting with coastal ecosystems, whether as a food supply or a source of recreation. Seafood is a food source with the potential to sustainably feed an ever-increasing population, and provide economic benefits, as the recent United Nations Food Systems Summit demonstrated [19]. The role of blue (or aquatic) food systems can play a significant role in establishing and maintaining food and nutrition security globally [20]. However, there are social challenges emerging from the human dimensions of the environment that serve as barriers to sustainably producing and distributing seafood. These barriers include, but are not limited to, the public trust, risk perception, and competing stakeholder interests at the local, regional, and global levels [21]. Human interaction within marine coastal ecosystems presents substantial risks to environmental sustainability, as well as interlinked food systems, with activities such as overfishing, and the contamination of marine ecosystems, impacting global food security, climate patterns, weather events, agricultural systems, and coastal flooding [22,23,24,25]. Additionally, marine ecosystems are influential in the livelihoods of many populations, and are thus a key area influencing multiple aspects of the SDGs [26], including SDG 8 “Decent Work and Economic Growth”, SDG 12 “Responsible Consumption and Production”, and SDG 14 “Life Below Water”.

In addition to the environmental consequences, marine ecosystem disruptions have aesthetic impacts on beaches, reduce recreational opportunities, and pose economic problems for community members near or within coastal ecosystems [22]. The public awareness surrounding the myriad economic, environmental, and social seafood industry impacts has encouraged industry players to begin adopting more sustainable standards for production [27]. Additionally, undergraduate students who, as part of Gen Z, are a burgeoning consumer market, are known as the sustainability generation, often changing their dietary patterns to those perceived to be more environmentally sustainable [28]. Thus, improving current and future consumers’ undergraduate education, and equipping them with the tools to think through the impacts of food choice on achieving the SDGs, may align with the emerging sustainability identities embedded within Gen Z (Gibson et al., 2023) [29,30]. With a growing public influence on the agenda of science, technology, and industry [2], students must have the skills to employ critical thinking with SSIs, to increase their confidence in solving problems and generating systemic solutions from an interdisciplinary perspective, to enhance sustainability across food system contexts [1,31,32,33].

The purpose of this study was to explore the use of HCSs in food system education contexts, in which students could learn systems-thinking content from a transformative-learning perspective. Three research questions guided the current study: (1) What were the responses provided by participants for each hypothetical case scenario? (2) How did students navigate the hypothetical case scenarios from a systems-thinking perspective? (3) How can the hypothetical case scenario method serve as a transformative-learning tool?

### 1.3. Theoretical Framework and Epistemological Perspective

The theoretical framework guiding the current study was transformative-learning theory [34], which emphasizes the shifts in attitudes, behaviors, and dispositions that can result from content-based learning. Within transformative-learning theory, learning is transformative when one uses critical reflection and awareness of their preconceived notions of a concept, restructures their assumptions about the concept, and begins to act according to those newly developed understandings of the concept [35]. According to Mezirow [34]:

Transformative learning refers to the process by which we transform our taken-for-granted frames of reference (meaning perspectives, habits of mind, mind-sets) to make them more inclusive, discriminating, open, emotionally capable of change, and reflective so that they may generate beliefs and opinions that will prove more true or justified to guide action. Transformative learning involves participation in *constructive discourse* to use the experience of others to assess reasons justifying these assumptions, and making an action decision based on the resulting insight.(p. 76, italics added)

Three elements are central to transformative-learning processes: experience, dialogue, and critical reflection [36]. Experience serves as the foundation for meaning-making and critical reflection, often through a disorientating dilemma: a “personal crisis, triggering event, or experience [that] challenges an individual’s belief structures” [37] (p. 358). Through dialogue, participants can be exposed to other experiences or perspectives that help them to critically question their underlying assumptions, which can act as barriers to sustainable solutions for complex issues [35,38]. Critical reflection continues the learning cycle, specifically through questioning one’s guiding assumptions, constructed through personal experience, perspectives, and beliefs [38]. The overarching goal of transformative-learning theory is facilitating change through a disruptive event, to help students, or other learners, critically examine their beliefs, followed by reconstructing beliefs, through dialogue and reflection, to beliefs that are more conducive to effective future actions [38].

While transformative-learning theory was conceptualized by Mezirow (1991) as an eleven-stage process, Boyer et al. [37] condensed the theory into a four-phase operationalization: (1) a disorientating dilemma occurs (2) followed by critical reflection, where an individual engages in “critical reflection and reevaluation of assumptions about themselves and learning”, (3) supported through validating discourse, in which an individual engages in “dialogue and discourse with other students or the instructor” (p. 358), and (4) emerging through reflective action, where an individual takes action across personal and professional contexts, resulting from their change in perspective (Figure 1). From an implementation standpoint, the instructor is a vital component to facilitating the critical reflection and dialogue, which are central to transformative learning [37].

Transformative learning has previously been implemented in relation to food systems and agricultural content (see [39,40]); however, specific learning tools for facilitating the transformative experience and incorporating systems thinking have been limited. Yukawa [38] explored the use of transformative-learning strategies in enhancing students’ capacity to deal with complex, wicked problems; or problems that occur in complex adaptive systems, in which causal relationships among actors and elements in a system are difficult to identify, due to their interdependent and interactive nature. Thus, there is a need to explore pedagogical strategies for facilitating transformative, systems-level thinking for students in the higher-education classroom, particularly within food systems contexts.

A key component of enacting transformative learning is to engage students in continual high-level reflection, in which they step back and assess what is occurring in the learning situation [38,41,42]. However, some scholars critique transformative-learning theory, citing a lack of attention to empathy, or the “ability to subjectively experience and share in another person’s psychological state or intrinsic feelings”, in order to foster transformative learning [36] (p. 43). Moral conflicts can arise within a transformative-learning context, due to potentially negative emotions from thinking in a systemic way, and there being negative consequences arising from certain aspects of the system [43]. Such previous observations indicate there is a current gap in praxis, and a need for instructional activities where students can engage in structured, safe dialogue, which can effectively engage, and manage, negative emotions arising from complex, morally charged, and potentially contentious discussions related to SSIs [3]. New research methodologies and pedagogical strategies are needed to explore transformative learning [36], and the current study aims to address this gap in the literature, by using HCSs in a transformative-learning context, using scenario-based learning to facilitate critical reflection, dialogue, and empathetic discourse through the construction and experience of exploring a HCS. The authors explore how HCSs can act as a learning tool to mitigate the cognitive dissonance arising from a disorientating dilemma, and how the use of structured dialogue from the instructor may help students arrive at a resolution for SSIs from a systems-thinking perspective.

## 2. Materials and Methods

A mixed-method research design was operationalized to address the research purpose and questions in the current study. The researchers selected an embedded mixed-methods approach, in which they first distributed a quantitative survey instrument with identical systems-thinking-based HCSs related to the food system, followed by focus-group sessions to debrief with students [44]. A mixed-method design was identified as the appropriate approach because of its ability to provide deeper knowledge of a subject, by employing methods that complement one another, enabling researchers to explore multiple facets of a phenomenon [44], while the mix of methods also allowed researchers to triangulate data, to lend trustworthiness to the research findings [45]. This study used non-probabilistic purposive sampling techniques to study the population of interest, which was undergraduate students with some familiarity with food systems. Participants in this study were purposively, non-randomly selected because of their status as undergraduate students enrolled in courses within the University of Georgia College of Agricultural and Environmental Sciences. The purposive sampling technique thus limited the generalizability of the study beyond the participants sampled. This study was part of a larger research effort to understand undergraduate students’ systems-thinking abilities and understanding and their food-system and environmentally conscious consumption values. The methods paralleled Sanders et al. [46], serving as an expansion of the pilot study and of the HCS methodology presented there.

### 2.1. Instrument Development

#### 2.1.1. Quantitative HCS Instrument

A quantitative survey instrument was distributed to participants through the Qualtrics online survey platform. The quantitative questionnaire contained three HCSs, a Likert-type systems-thinking scale [47], a green consumption values scale [48], and demographic questions. For the purposes of this study, only the HCS answer selections and demographic responses were used. When participants logged into the questionnaire, they were presented with the first of three systems-thinking archetypal (see [49]) HCSs. After completing the HCSs (detailed below), participants were presented with the Likert-type systems-thinking scale [47], on which participants were asked to identify how often, in their personal process for making improvements, they participated in a series of 20 items (0 = Never; 4 = Most of the time). The responses to each item were summed to create a systems-thinking score from 0 to 80 [47]. On the next page, participants were asked to indicate their level of agreement on a seven-point GREEN scale inquiring about their level of agreement or disagreement with six items related to their environmentally friendly consumption values (1 = Strongly disagree; 7 = Strongly agree). the responses to the GREEN scale items were averaged, forming a single green consumer values score [48]. Following their completion of the HCSs and scales, participants answered demographic questions. Participants were asked to self-identify their age, gender identity, race/ethnicity, college of enrollment within the University of Georgia, classification as a student by year (first-year, sophomore, and so forth), the rurality of the community in which they grew up (rural, suburban, or urban), if they grew up in a coastal community, and any dietary restrictions.

In each HCS, the participant was presented with a primary scenario, and asked to choose between two options to address the scenario described. Each participant made their primary choice, then they were directed to one of two possible branched primary outcomes. The two primary outcomes provided a description of the events that took place because of the primary choice, and subsequently presented them with two secondary choices. Once the participant selected a secondary choice, they received a description of their final scenario outcome. Because two sets of secondary outcomes were available per primary outcome, there were a total of four possible final scenario outcomes for each HCS. A graphical representation of the learner process is presented in Figure 2.

The three scenarios were presented to individual participants in a random order within the survey instrument, to reduce potential bias, and to mitigate the potential influence of one scenario on another throughout the survey process.

The first HCS presented a case detailing select issues surrounding Atlantic cod decline and its effect on coastal communities, drawing upon details from real-life cases of this issue [50,51,52]. Researchers modeled the first HCS around the “tragedy of the commons” archetype that frequently arises in systems-thinking analysis, as defined by Meadows [49]. The second HCS dealt with an oyster operation looking to expand, but faced with potential detrimental environmental impacts if they remained in their current location. Employing Meadows’ [49] “fixes that backfire” archetype, this HCS also contained details from actual events reported in research or the popular press [53,54,55,56]. The third and final HCS; modeled using the principles of Meadows’ [49] “accidental adversaries” archetype, and actual events [57,58,59,60,61]; presented a case in which a community-planning commission was faced with choosing between investment in the local tourism industry or the local fishing industry.

Before the survey distribution, a panel of five experts with professional and academic experience in natural resources, aquaculture, agricultural communication, agricultural leadership, agricultural education, extension education, and quantitative methods reviewed the instrument for face and content validity. The focus-group protocol (detailed below) was reviewed for both face and content validity by three university faculty members with expertise in agricultural communication, agricultural leadership, and program evaluation. A pilot test was conducted with a group of graduate students in the University of Georgia agricultural communication course, for both the quantitative and qualitative instruments. Researchers adjusted the protocol according to feedback from both the experts and the graduate students.

#### 2.1.2. Qualitative Focus Groups

Researchers developed a focus-group protocol to provide participants with the chance to reflect upon, and deliberate with peers, the various outcomes of each scenario they encountered in the quantitative HCS instrument. Questions within the focus-group protocol were each associated with one HCS contained within the quantitative HCS instrument. A moderator provided a brief definition of the systems-thinking archetype that participants saw in the HCS instrument, before beginning the discussion of each individual scenario. The moderator asked participants to raise their hands to indicate the final outcome they received as a result of their selections in the scenario at hand. Participants were subsequently asked to describe how they went about making their decisions within the HCS. Once participants had the opportunity to discuss this question with their peers, the moderator distributed a physical copy of the scenarios to show participants the different possibilities available as primary and secondary outcomes. One minute after receiving the handout, the participants were asked: (1) what they expected to happen, (2) how those expectations differed from what played out in the scenario, and (3) now that they have seen the whole scenario, would they make different choices. This process was repeated for each scenario. After all scenarios were discussed, the moderator asked participants to reflect upon and summarize in one sentence their perception of the experience with the quantitative HCS instrument, and the focus-group session. To ensure that each participant had an opportunity to respond to this question, the moderator called each participant by name, to provide a turn for their response.

### 2.2. Data Collection and Analysis

Data were collected from 10 November 2021 to 20 April 2022, amongst students enrolled in five courses in the College of Agricultural and Environmental Sciences. The students were advised as to the study’s voluntary nature, and that declining participation would have no effect on their class grade or standing; students voluntarily signed consent forms to participate. Data collection was completed during regularly scheduled class times; however, the primary instructor was not present. All data were collected by the research team, none of whom had existing relationships with participants. A total of 68 students completed both the survey and focus-group portions of the study. Participation was limited to only those individuals who were in class during the data-collection process. All participants elected to participate in the study. The current study was approved and deemed exempt by the University of Georgia Institutional Review Board (Protocol #00004479).

When participants arrived in the classroom, researchers asked participants to sign in using an identification number sheet. Participants were then randomly assigned into focus groups, with the goal of 8 to 10 participants per group. First, researchers presented a short overview of what the focus-group sessions would entail, and then they provided participants with a sheet of paper that contained a unique identification number, to ensure participant anonymity in the survey instrument, a room assignment (if needed), and a section for documenting their results associated with each HCS. The provided sheet was used for students to note their anonymous identification number when completing the survey instrument, and for their personal reference during the focus group. Following the dissemination of the identification sheets, the focus groups followed their moderators into designated rooms to begin their individual sessions. Each focus-group room contained 8 to 10 participants. A total of 12 focus groups were conducted across the recruited classes. Students within each focus-group session were given time to complete the Qualtrics survey instrument, and record any reference information about their results. Following the survey, participants engaged in focus-group sessions, the qualitative portion of the mixed-method study.

Designed to understand individual and group experiences and opinions about a specific phenomenon through group discussion, focus groups utilize group interaction to gather data [62]. During focus groups, participants listen to the responses of others, and expand on comments, to extend the discussion beyond the original responses. This allows the opportunity for participants to express their individual opinions, while considering the opinions of others [63] (Patton, 2014). On average, focus-group sessions took 45–60 min each. When the participant attendance was large enough, concurrent focus-group sessions were conducted during the class period. Each focus-group session had at least one notetaker and one moderator. All moderators used the same moderator guide throughout the study, and in all focus groups.

The focus-group data were transcribed verbatim and then uploaded to MAXQDA qualitative analysis software for analysis. Researchers used an inductive coding approach, and the constant comparative method outlined by Glaser and Strauss [64]. Two authors coded the data for the overall gestalt value, followed by axial coding to generate themes connecting across the 12 focus groups [65]. Instead of allowing codes to be identified from pre-determined theory, the themes were identified within the data in vivo [65]. The two authors generated a codebook to document both the development of, and agreement between, codes [65], followed by peer debriefing with the entire research team, to enhance the data trustworthiness [45].

In accordance with recommendations within the literature [66], a subjectivity statement is provided to assist in the interpretation of the results. At the time of data analysis, the primary coder was a third-year doctoral candidate studying science communication in the college in which the focus groups were conducted. With a focus in science communication and program evaluation, she specializes in identity-informed communication and evaluation, and has an emerging interest in the influence of systems thinking on agricultural and environmental innovations. The second coder was also a science communication doctoral student, with a focus in science communication, enrolled in the college in which the focus-group sessions were facilitated. She possesses an undergraduate degree and master’s degree from the same college, and has professional experience in agriculture, as well as experience in international and emerging aquacultural industries, which may have influenced her views on food systems. Additionally, her familiarity with undergraduate student life in the college may have influenced her perception of student identities within the focus groups; however, she did not know any of the focus-group participants prior to the sessions. The frequencies and percentages were operationalized, to describe participant demographics (see Table 1), and descriptively analyze the quantitative HCS survey data, using SPSS 26.

## 3. Results

### 3.1. Responses to the Seafood Hypothetical Case Scenarios

The frequencies for the Atlantic cod scenario, using the “tragedy of the commons” archetype, indicated that a majority of participants prioritized sustainability (n = 54) over fishing for food (n = 14) at the first choice stage. Based on the four potential outcomes in the secondary choice stage, most participants (n = 39) selected lobbying for governmental credit systems as their preferred intervention within the HCS (see Figure 3).

The second scenario focused on oyster farming, and used the “fixes that backfire” archetype. This scenario illuminated participants’ decisions related to moving to a new location, or expanding in the same location, an oyster-farming operation facing environmental and economic impacts. Initially, most participants (n = 47) chose to move their operation and face potential economic costs, in order to prioritize the environment. At the second choice stage, most participants decided to prioritize their product quality by purchasing expensive land primed for oyster farming (n = 43), considering both the long- and short-term economic impacts of their decision (see Figure 4).

The last scenario highlighted a local community commission, and used the “accidental adversaries” archetype (see Figure 5), presenting a decision between investing in local tourism or in the local fishing industry. Initially, more participants chose to invest in local tourism (n = 39) than in local fishing operations (n = 29). At the second choice stage, many participants who chose tourism investment elected to invest in local tour-guide training and tourism services (n = 24), while the majority of those who chose local fishing operations at the first stage elected investment into local conservation efforts (n = 27).

Research questions two and three were qualitative in nature. Table 2 provides an outline of themes and subthemes identified within each research questions. Themes and subthemes are discussed in detail in the proceeding sections.

### 3.2. Navigating the Hypothetical Case Scenarios from a Systems-Thinking Perspective

Five themes were identified related to systems thinking in participants’ responses: (1) methods for reasoning through scenarios, (2) surprise and frustration at outcomes, (3) understanding of, and confidence in, choices, (4) evaluating systems-level consequences, and (5) thinking through systems complexity.

Students used various methods to reason through the scenarios, one being thinking in long-term rather than short-term mental frameworks. For example, one student in focus group (FG) 9 explained their reasoning for the “tragedy of the commons” scenario:

I chose prioritizing sustainability because in the long run that would last longer than prioritizing feeding the population. If you fed them all too much at once then you could run out quicker. But if you did sustainability, it could last a little bit longer.

Another student explained, “compared to the other [choices], that’s probably the better choice maybe even though you would take the upfront like relocation cost, I think it would pay off in the long run and you wouldn’t hurt the local environment” (FG5). Other students made decisions based on the appropriateness of the solution for the audience: “I […] went the fishing limit route [for the “tragedy of the commons” scenario], because not all of these fishermen are going to have social media to view the campaign that they’re trying to get advice” (FG11). Another student reflected on the appropriateness of the decision for stakeholders, saying:

[When] choosing between social media or government intervention, everyone doesn’t have social media. Social media is just [for] a certain generation, or everyone is on different social media sites [….] I feel like it would be hard to reach people with a social media campaign over the government.(FG8)

Finally, several students described their decision-making process as trying to minimize negative outcomes within each HCS. For the “accidental adversaries” scenario, one student in FG5 said, “I expected there to be more traffic with the environment. I know it would cause a little bit damage maybe not as much as it as is explained here”.

Participating students also expressed surprise and frustration at the outcomes of the scenarios. Several students expressed surprise at there being no correct, or “winning” (FG2), option. Moving through the scenarios, students realized that “every scenario has its downside—that’s what surprised me” (FG2). Other students were frustrated with this non-linearity, saying, “[in the “accidental adversaries” scenario,] all the choices make someone angry and you always lose your campaign, there’s no win here” (FG5). Some students experienced unmet expectations in their decision-making:

I [would] still choose what I chose, but I’m sad when you know what happened. I made those decisions. I thought that sustainability will always work through and it’s not just planning ahead, because it will work out, but it doesn’t work out.(FG11)

Other students stated that, knowing what they knew now about the results, they would change their choice: “Knowing all of the answers now, I’d rather have a nice environment than to have another [touristy] beach somewhere, so that’s what I would’ve done” (FG10). The scenarios provided reflective opportunities to discuss unintended consequences from decisions after the systems-thinking debrief. One student reflected:

When I prioritized environmental sustainability, I thought that it was going to be the right choice, or it was going to have a good ending. But [… my results] just showed me how […] things that you won’t plan for also go wrong. [In the fixes that backfire scenario…] it’s more than that, it’s consider[ing more] than just a typical, one size fits all type situation to cover up the issue of at hand.(FG9)

Within the third theme, understanding of, and confidence in, choices, participants had various reactions to navigating the decision-making process, leading to different learning outcomes about their own ways of thinking. Two subthemes were identified, which categorized participant responses: justification of choice, and comparing outcomes. Participating students provided several examples of the justification of their choices. Related to the oyster-farming scenario, a participant outlined their decision process:

I chose to remain in the same location on that first choice because the conservation green lighted it. I figured I’d go ahead and try it and until something new arose instead of trying to preemptively change my mind. Then when something did arise, I chose to go ahead and cut back on those detrimental actions to preserve the environment. I had pretty good outcomes with [that choice].(FG11)

Another student explained, related to the “tragedy of the commons” scenario, “I was evaluating the other option, but I would still stick to sustainability […] just because social media doesn’t really resonate with me” (FG2). Other students were confident in their choices, and justified their decision through comparing the various outcomes. A student in FG8 stated:

I would stick on that path. I [figured] I would face more cost in the beginning, but that’s a risk that you have to take in a business anyway, and then making those more expensive choices set me up for success in the long run to pay it back and then profit off it.

Participating students engaged in different techniques to evaluate systems-level consequences, the first being comparing hypothetical with real-world challenges. One student explained, “we thought we had unlimited money […] So in a mythical world, yes, I would relocate, but in the real world who knows what would’ve been my best option in terms of finances and everything” (FG10). Other students evaluated tradeoffs between economic prosperity and environmental sustainability: “it’s difficult to realize both environmentally sustainable fishing practices and doing beneficial business” (FG10). Another way in which students explored systems-level consequences was through evaluating the various risks associated with each option in the HCSs. A student explained their reasoning through the decision-making process:

I [thought,] I’m going to just take the risk of relocating. When it came to the other options, I’d much rather pick high quality over something lower […] So I just want what’s best and then I’m going to assume the consequences of my actions […] It’s just a risk of getting to a higher level, there’s always going to be a risk.(FG2)

Overall, participants chose to prioritize sustainability, economics, or the community in their decision-making process. Some students relied on their backgrounds and previous experiences to make their choice:

I came from a really small town. The people that ran for office were people who grew up there. They wanted the best for the community. I lost reelection but it doesn’t bother me because I did what I thought was best for the community. (FG6)

Other students prioritized sustainability, stating “I opted for the environmentally safe one” (FG6), “I tried to keep sustainability in mind when making those choices” (FG7), and “I wish I had chosen to focus on sustainability […] since the environment’s damage was permanent” (FG4). Evidence of participants prioritizing economics included statements such as “I was just thinking in terms of money” (FG7), “I chose to purchase more expensive land. I kind of figured obviously lose profit, but if you purchase more expensive land and just work harder, you’ll make your money back” (FG2), and “I’d rather have top quality oysters and sell them, make a profit on those instead of giving lower quality oysters, which my business can again suffer” (FG6).

Finally, the fifth theme was thinking through systems complexity, where students imagined improvements to the scenarios, complexified potential solutions, and recognized the unintended consequences of decisions. Recognizing the inherent limitations of only having four choices, participating students dialogued about what they would change to make their result more suitable for a variety of stakeholders:

I feel like something […] that might be interesting is if we wind up with result [increasing investments to local conservation efforts (Figure 3)], there’s room for fishermen to move into tourism. There’s still room to have your own business and make your own money. Maybe you start charging fishing boats [tours …] or something. That’s not really named here but it’s a potential outcome for those fishermen, a potential business.(FG11)

Another student explored a third possibility not presented in the scenarios: “Out of those two choices, there could’ve been a third choice that might’ve been a better option, like placing a temporary ban on the Atlantic cod or just making it a little harder to fish [through other] regulations” (FG1).

Related to unintended consequences, one student reflected, “in the end, everything will have a downside. Whatever decision we make, so you just want to do good, do what’s best, what you think is best” (FG2). Another student mused, “I realize that sometimes a good heart and good intentions, don’t always have positive consequences, and [that] made you realize how many unintended consequences were coming from every action that we ‘chose’” (FG9). A student reflected, “most decisions often have unpredictable outcomes” (FG3).

### 3.3. Hypothetical Case Scenarios as a Transformative-Learning Tool

Four themes were identified to provide evidence for using HCSs in a transformative-learning context: using previous knowledge and experience, experiencing difficulties and confusion, developing new ways of thinking, and experiencing positive learning outcomes. Several students described how they used their previous experiences and existing knowledge to help them chose an option in the HCS. One student explained,

I think [my choice] has to do with the fact that I grew up in Las Vegas. I was immersed into tourism culture. In my head it didn’t bring benefit. I saw a lot of the bad parts of it. So, I just figured community-based [industry] might be better.(FG6)

Other students experienced difficulties and confusion in the decision-making process, emulating a disorientating dilemma. Often, the disorientation came from pedagogical strategies that were new to students: “I feel just because we are in school I expect there to be a ‘right’ answer, so sometimes it’s a little frustrating because you’re trying to get to the right answer and everything’s wrong” (FG7).

Other participating students described how the HCSs helped them to develop new ways of thinking, or to think about their cognitive processes in a different manner. Many reflections came from the perceived differences between the real world and what is learned in the classroom: “[this activity] was very helpful to see that, our actions could have drastic consequences. There’s rarely a right or correct answer when it comes to our choices in life outside of a classroom” (FG6). Another student explained what they learned about deficits in their own thinking: “I learned I did not realize the trickle-down effect of potential choices” (FG10).

Overall, participating students felt the HCS activity had positive educational outcomes. When asked to describe the HCS experience, one student stated, “I feel like it was educational and I really like how this study is discussion-based because you’re able to see other people’s choices and how their backgrounds can affect their choice-making” (FG6). Another student described the activity as “very thought-provoking and informative” (FG3). The discussion-based setting of the HCS activity was a key component appreciated by participating students:

I think it was interesting just to get to hear everyone’s perspective and like opinions, because I think it really emphasizes on the fact that your opinions are built [out] of where you come from, who your family is, where you live, and [your] background. So I thought that was interesting to hear about and just to see everyone’s perspective.(FG2)

## 4. Discussion

Overall, the results indicated that the HCS activity was an impactful learning tool for increasing students’ capacity to engage in systems thinking around SSIs, particularly related to food-system scenarios. Participants in all 12 focus groups were highly involved in the discussions and debriefs of the HCSs, leading to generative dialogue and exposure to others’ perspectives and experiences. For many students, the HCSs were also their first time exposed to educational content around seafood and costal industries. The HCSs may then serve as an engaging strategy for introducing new, complex scientific and food-system topics into the classroom, especially those related to the SDGs.

The quantitative analysis of the HCS choices revealed that higher numbers of participants were likely to choose environmental sustainability, as exhibited by the popularity of the first choices leaning toward environmental sustainability in both the Atlantic cod scenario (“tragedy of the commons”) and the oyster-farming scenario (“fixes that backfire”). In the “accidental adversaries” scenario, participants were more evenly split between the tourism and fishing industries, neither of which were inherently environmentally sustainable. However, the qualitative analysis of the focus-group reflections emphasized the complexity of students’ decision-making processes, and allowed them to exhibit knowledge of complex topics beyond that of environmental sustainability, even when the content area was unfamiliar to many participants.

Participating students exemplified components of systems thinking in their decision making, specifically evaluating tradeoffs between the three legs of sustainability (economic, social, and environmental) [6], as well as reflecting on the unintended consequences that arose as a result of those decisions. Additionally, the HCSs can potentially address a gap in current transformative-learning theories, specifically criticisms stating that the theory excludes attention to empathy. HCSs, through the choose-your-own-adventure-type setting, as well as through group dialogue, expose students to others’ experiences, helping them to “subjectively experience and share in another person’s psychological state or intrinsic feelings” [36] (p. 43). Through the moral conflicts and frustration arising from the HCSs, students can engage in structured, safe dialogue, which enhances holistic learning outcomes related to the SSIs and SDGs.

Despite positive results, there are limitations associated with the current study, which should be acknowledged. Firstly, due to the authors’ need to use an educational setting as an intervention, they relied on convenience samples of students enrolled in courses within the college of agriculture to recruit participants. Thus, findings are exploratory, and should not be generalized beyond the sample. Additionally, participants may not have been a representative sample, due to the non-random sampling techniques used; however, the participant demographics were provided, to enhance the transparency of the sample. An additional limitation was that the HCSs were implemented as a one-time intervention in each of the five classes in which participants were enrolled, and thus limited to a single point in time, as opposed to an ongoing learning approach used throughout the duration of the course. However, the results from the single intervention suggest that HCSs have the potential to foster systems thinking, especially from a transformative perspective, by introducing a disorienting dilemma, encouraging dialogue, and constructing reflection about metacognition. A recommendation would be for future studies to examine the use of HCSs to explore students’ systems thinking and transformative learning over a longer duration, replicating not only the HCSs presented here, but also adapting HCSs to other food-system topics. The measures of impact could incorporate both quantitative scales to measure systems-thinking capacity, and reflection and group dialogue. Another consideration is that, despite the HCSs and systems thinking being interdisciplinary in nature, the majority of participants came from majors defined as disciplinary. Thus, participating students may not have been introduced to interdisciplinary work, such as systems thinking, prior to the educational intervention. Alternatively, there was not a measure of participants’ previous exposure to interdisciplinary work at the undergraduate level. Future research should explore how the role of interdisciplinarity in participants’ backgrounds could impact their experience of the HCS intervention. Finally, class periods were limited to 45 to 60 min, which limited the focus-group debrief and dialogue that occurred. The implementation of the full HCS process is recommended to be a minimum of one hour and a half, to account for set up, the HCS activity, and debrief, to ensure that all students have the opportunity to fully participate.

Building from Mezirow [35] and Boyer et al. [37], the quantitative and qualitative findings of the current study were consolidated, to propose a model for implementing food-system HCSs through a transformative-learning perspective (Figure 6). Most students expressed both surprise and frustration at the outcomes of their scenario decisions, which related to Mezirow’s [35] concept of a disorientating dilemma. Additionally, students built on their previous knowledge and experiences as a basis for new learning, to contribute to generative dialogue and group reflection. The instructor, or educational facilitator, played a crucial role in conceptualizing the transformative-learning experience into new learning outcomes that students could carry with them beyond the classroom. Figure 6 outlines the process for implementing HCSs in a higher-education classroom.

The process initiates with a student as an individual, able to build upon and reconstruct their preconceived notions of a topic. The second stage contains the disorientating dilemma; in this case, the HCSs as a learning tool to introduce cognitive dissonance between the student’s preconceived notions and the unintended outcomes of a systems perspective for problem solving. After the student experiences the disorientating dilemma, the instructor facilitates learning and concept reconstruction, by introducing systems-thinking concepts related to the HCS learning tool. This stage acts as a debrief and foundation for student reflection on the HCSs themselves. The fourth stage contains collaborative reflection, which occurs during the debrief, between the individual and the group. This stage allows for the collective reconstruction of assumptions and concepts related to the HCSs. Finally, the cycle terminates, and reinitiates back at the first stage, where the student integrates the new reconstruction of the concept, and connects to other related notions within and beyond the process.

## 5. Conclusions

HCSs, as a transformative teaching tool, have the potential to leverage SSI education, specifically relating to complex issues, including those related to food systems. Now, and into the foreseeable future, science and food educators have a responsibility to contribute to a sustainable future. Through educational strategies such as HCSs, educators can help students to contextualize human actions as being integrated within the environment, and to better understand how their actions and decisions can have impacts well beyond the proximal cause and effect observed within a finite situation [5]. Future research should examine HCSs over a longitudinal period of undergraduate studies, with baseline measurements of systems-thinking capacity [47], to determine the quantitative impact of such experiential learning techniques on students’ systems-thinking development. Additionally, the authors recommend replicating the methods presented here, and in Sanders et al. [46], across different context areas, to isolate the impacts of systems-thinking HCSs on student development. Specifically, replications could enhance content areas related to the SDGs, and education promoting the sustainable development of blue food systems to enhance global food security.

## Figures and Tables

**Figure 1 foods-12-02663-f001:**
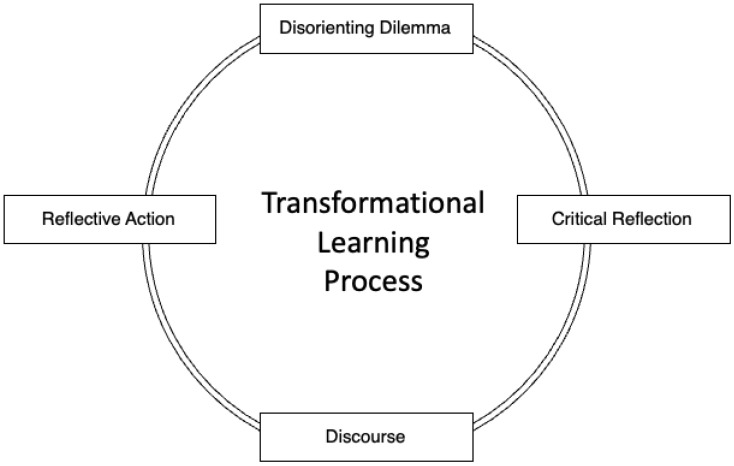
Transformational learning process. Adapted from Boyer et al. (2006) [37].

**Figure 2 foods-12-02663-f002:**
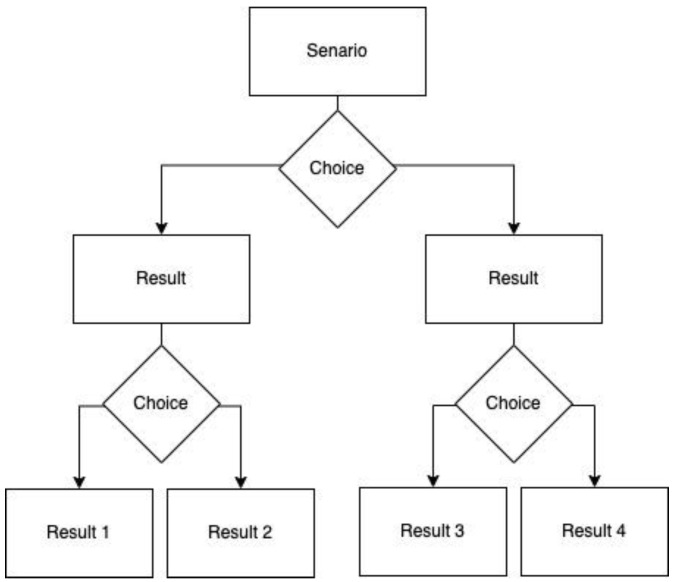
HCS decision tree graphical representation.

**Figure 3 foods-12-02663-f003:**
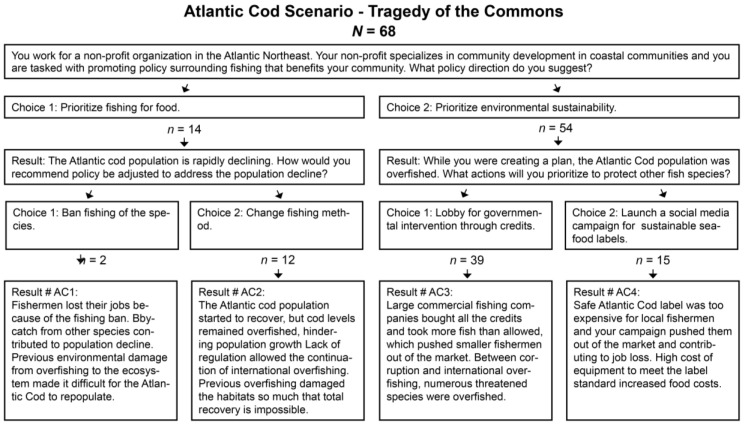
Adapted from Brodwin [50], Cudmore [51], and Food & Water Watch [52].

**Figure 4 foods-12-02663-f004:**
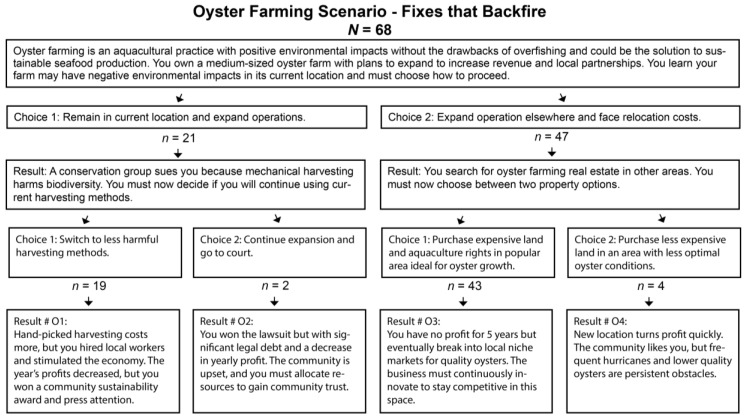
Adapted from Kraft [53], Petrolia and Walton [54], Sink et al. [55], and Tallis et al. [56].

**Figure 5 foods-12-02663-f005:**
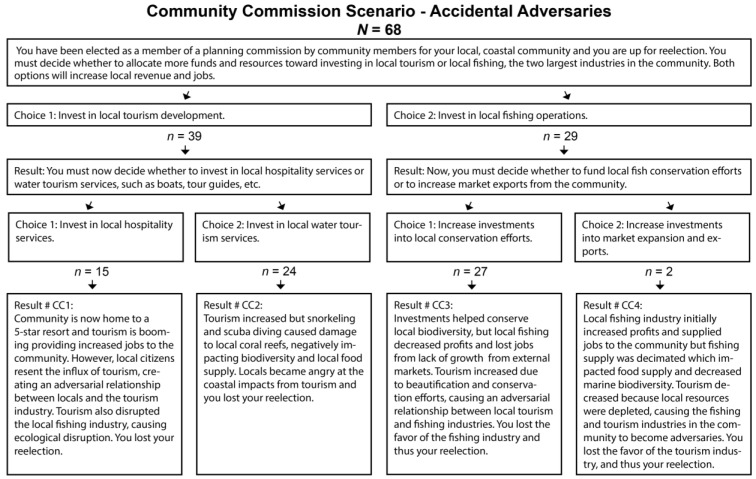
Adapted from Baynes [57], Coral Reef Alliance [58], Dance [59], National Ocean Service [60], and Sustainable Travel International [61].

**Figure 6 foods-12-02663-f006:**
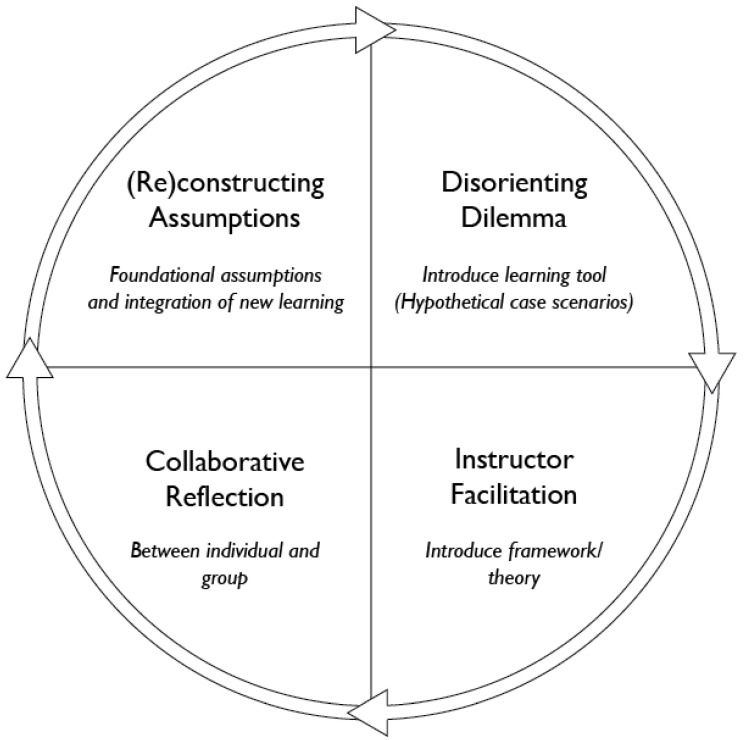
Revised model for implementing food system hypothetical case scenarios, expanding on Boyer et al. [37].

**Table 1 foods-12-02663-t001:** Participant demographics.

	*F*	*%*
Gender Identity		
Male	20	29.4
Female	47	69.1
Age		
18	3	4.4
19	15	22.1
20	14	20.6
21	16	23.5
22	13	19.1
23	6	8.8
24	1	1.5
Race/Ethnicity ^a^		
White	55	80.9
Black or African American	8	11.8
Asian	2	2.9
Hispanic or Latino/a/x	4	5.9
Prefer to self-describe (Middle Eastern)	1	1.5
Student Classification		
First-year student	4	5.9
Sophomore	12	17.6
Junior	25	36.8
Senior	24	35.3
Graduate student	1	1.5
Other (Exchange student)	1	1.5
College Enrollment		
College of Agricultural and Environmental Sciences	44	64.7
College of Arts and Sciences	9	13.2
College of Business	6	8.8
School of Public and International Affairs	2	2.9
College of Engineering	1	1.5
College of Journalism and Mass Communication	1	1.5
College of Education	1	1.5
Undeclared	1	1.5

Note: ^a^ Respondents could select more than one item.

**Table 2 foods-12-02663-t002:** Research Questions and Related Themes and Subthemes.

Research Question	Theme	Subtheme
Navigating hypothetical case scenarios from a systems-thinking perspective	Methods for reasoning through scenarios	Long-term mental frameworks
		Appropriateness of solution for audience
		Minimizing negative outcomes
	Surprise and frustration	No correct answer
		Unmet expectations
		Surprise at negative outcomes
		Switching choice
	Understanding of, and confidence in, choice	Justification of choice
		Comparing outcomes
	Evaluating systems consequences	Prioritizing sustainability
		Prioritizing economics
		Prioritizing community
	Thinking through systems complexity	Imagining improvements
		Recognizing unintended consequences
Hypothetical case scenarios as a transformative-learning tool	Using previous knowledge and experiences	
	Experiencing difficulties and confusion	
	Developing new ways of thinking	
	Experiencing positive educational outcomes	

## Data Availability

The data presented in this study are available on request from the corresponding author. The data are not publicly available due to participant confidentiality needs.

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
