# Peer review of "Teaching Systems-Thinking Concepts with Hypothetical Case Scenarios: An Exploration in Food-Systems Science Education"

_foods, 2023, doi:10.3390/foods12142663_

Round 1

Reviewer 1 Report

This review's primary aim is to give some pointers to the paper authors entitled “teaching systems thinking concepts with hypothetical case scenarios: an explanation in food systems science education”. In my view, the paper weakness outrun the strengths in that it is poorly organised and it is far from being precise. To be more specific, the drawbacks are as follows:

(1) Of the shortcomings, one of the most severe is how poorly organised the paper is. Although the introduction succeeds in addressing the subject (lines 108-110), justifying its importance (lines 65-101) and spotting the research gap (lines 102-107), it is devoid of giving and advancement of the paper structure and setting out the paper research objectives. So, please, append to the introduction the paper structure and set out the research objectives.

(2)   The literature review needs more direction, so its content seems unpredictable (lines 111-178). To overcome this drawback, I suggest that you define the key concepts at the beginning and next, you explain the model to test. Let me recommend that you make good use of figures and illustrations. It is desirable to answer the research questions and you might divide the review of the literature section into as many subsections and subheadings as research questions to answer.

(3)   I believe that a sensible content distribution moves the meaning along and there is an order that aids comprehension. Undoubtedly, purpose and research questions should be placed at the start instead of being in the review of the literature section (lines 180-187). For this reason, I suggest you move the purpose and research questions to the introduction instead of leaving them in the literature review.

(4)   Nothing against using several research techniques. However, this miscellaneous approach should be clear and organised. Let me suggest that you are more systematic by providing the reader with more methodical, accurate and exhaustive information about the survey and focus group. For example, it is advisable to give information about the survey context, the sampling procedure (probabilistic or non-probabilistic), the population, the sample profile and the sampling unit. Moreover, it is worth noting that you reflect on how representative your sample is.

(5) Regarding the methodological section, the theoretical contents are out of place. Lines and paragraphs are full of bibliographical references (lines 202-228). It is not the educational context that matters most but rather the research context. In other words, when, who and where did you apply the research techniques? Please, delete doctrinal contents from the methodological section and describe the research context techniques here instead.

(6)   The measuring instruments used in the questionnaire are presented (lines 231-244), but nothing is said about the questionnaire structure. Please, describe the questionnaire structure and the bibliographical sources you use to build the scales.

(7)   You touch upon the focus group, but something needs to be said about the raised questions and the research context (lines 277-293). Please, tackle it.

(8)   On no account does the methodological section represent 50% of the paper's full extension. It is so long that even the methodological section reaches ten pages (lines 188-372)! You indeed provide information about the participant's profile, but reading it is tiring too because all this is in the text rather than in tables. Please, cut down on it and be more efficient.

(9)   You culminate the paper with a conclusion section, but it needs future lines of research. So, please, give future lines of research.

(10)Not only is the paper devoid of a discussion section, but there are many doctrinal dispersed contents. Therefore, I would like to know if you might create a discussion section without making the paper longer than it is now.

These comments help improve their paper and encourage you to progress.

Author Response

(1) Of the shortcomings, one of the most severe is how poorly organised the paper is. Although the introduction succeeds in addressing the subject (lines 108-110), justifying its importance (lines 65-101) and spotting the research gap (lines 102-107), it is devoid of giving and advancement of the paper structure and setting out the paper research objectives. So, please, append to the introduction the paper structure and set out the research objectives.

  • ACTION: We have revised the structure of the paper to better set out the research questions/objectives.

(2)   The literature review needs more direction, so its content seems unpredictable (lines 111-178). To overcome this drawback, I suggest that you define the key concepts at the beginning and next, you explain the model to test. Let me recommend that you make good use of figures and illustrations. It is desirable to answer the research questions and you might divide the review of the literature section into as many subsections and subheadings as research questions to answer.

  • ACTION: We have attempted to clarify the literature review with subsections as well as added figures.

(3)   I believe that a sensible content distribution moves the meaning along and there is an order that aids comprehension. Undoubtedly, purpose and research questions should be placed at the start instead of being in the review of the literature section (lines 180-187). For this reason, I suggest you move the purpose and research questions to the introduction instead of leaving them in the literature review.

  • ACTION: We have moved this section to the introduction.

(4)   Nothing against using several research techniques. However, this miscellaneous approach should be clear and organised. Let me suggest that you are more systematic by providing the reader with more methodical, accurate and exhaustive information about the survey and focus group. For example, it is advisable to give information about the survey context, the sampling procedure (probabilistic or non-probabilistic), the population, the sample profile and the sampling unit. Moreover, it is worth noting that you reflect on how representative your sample is.

  • ACTION: We have added that the participants were recruited through purposive, non-random sampling methods and added that this limits the generalizability of the study. We also added considerations for the representativeness of the sample in the discussion section.

(5) Regarding the methodological section, the theoretical contents are out of place. Lines and paragraphs are full of bibliographical references (lines 202-228). It is not the educational context that matters most but rather the research context. In other words, when, who and where did you apply the research techniques? Please, delete doctrinal contents from the methodological section and describe the research context techniques here instead.

  • ACTION: Removed theoretical contents from the methodological section and moved this to the introduction where it is more fitting.

(6)   The measuring instruments used in the questionnaire are presented (lines 231-244), but nothing is said about the questionnaire structure. Please, describe the questionnaire structure and the bibliographical sources you use to build the scales.

  • ACTION: Moved questionnaire structure information and added additional information about questionnaire structure, as well as bibliographical sources used for the Likert-type scales used in the survey.

(7)   You touch upon the focus group, but something needs to be said about the raised questions and the research context (lines 277-293). Please, tackle it.

  • ACTION: We have revised this section to clarify the research methods used to guide the focus group protocol. We hope our revisions have addressed the reviewer’s concern, though we are happy to further revise with subsequent guidance.

(8)   On no account does the methodological section represent 50% of the paper's full extension. It is so long that even the methodological section reaches ten pages (lines 188-372)! You indeed provide information about the participant's profile, but reading it is tiring too because all this is in the text rather than in tables. Please, cut down on it and be more efficient.

  • ACTION: Changed description paragraph of participant demographics to a table for easier readability

(9)   You culminate the paper with a conclusion section, but it needs future lines of research. So, please, give future lines of research.

  • ACTION: We have added future lines of research.

(10)Not only is the paper devoid of a discussion section, but there are many doctrinal dispersed contents. Therefore, I would like to know if you might create a discussion section without making the paper longer than it is now.

  • ACTION: We have revised the structure of the discussion and conclusion sections.

Reviewer 2 Report

Dear authors,

In this game-like study, the authors propose hypothetical case scenarios (HCS) as a tool for systems-level learning. They explore the decisions and outcomes of 68 students on three HCSs related to the seafood system. The HCSs were designed on socio-scientific issues based on the tradeoff between sustainability, food production, and socioeconomic outcomes. The theoretical and epistemological approaches are based on the transformative learning theory, which served as background to design the method used in the study. The method is very well described, presenting the various steps and detailing the procedures in each step. Results of choices done by participants regarding each of the HCSs (Atlantic Cod Scenario/Tragedy of the Commons, Oyster Farming Scenario/Fixes that Backfire, and Community Commission Scenario/Accidental Adversaries) are quantitively and separately presented. Additionally, the authors report results exploring the participants' perceptions regarding reasoning through scenarios, surprises and frustrations, choice process, consequences, systems complexity, and the HCSs as a learning tool in a qualitative manner. In the conclusion section, the authors provide a model for implementing HCSs in food systems and conclude that HCS can qualify the education on socio-scientific issues related to complex systems, including food systems, with the potential to be applied to other food systems.

The article deals with a relevant topic aligned with the scope of the journal Foods. The complexity of food systems is continuously increasing, and social-scientific issues have become tradeoffs that confuse individuals' decision-making processes and choices. The proposition of a tool that enables reflection and learning related to these complex tradeoffs is welcome. Applying such a learning tool at the undergraduate level is timely to expand the possibilities of choices and realize the consequences of decisions. The outcomes are not always predictable and positive, and learning how to deal with this is necessary.

In addition, the manuscript presents the scientific attributes that underlie the theoretical rationality of the proposed method. Finally, the method is presented straightforwardly, allowing its replication.

Just a comment to the authors reflecting on and, eventually, addressing some considerations in the text: the participants in the study were students enrolled in Agricultural and Environmental Sciences, Arts and Sciences, Business, Public and International Affairs, Engineering, Journalism and Mass Communication, and Education. Although the group of participants has some multidisciplinary backgrounds, each participant's particular background is disciplinary in nature. The analysis of complex and systemic issues can benefit from an interdisciplinary background. How could an interdisciplinary background affect the results? Was there any concern about this topic during the definition of the study design? How could interdisciplinarity be considered in future studies applying the HCS tool?

Author Response

This concern was not addressed in the study design as the collaborative nature of the HCS in the classrooms were multidisciplinary due to students’ backgrounds. However, we have added a limitation and considerations for future research regarding this issue to the conclusions section. We appreciate this opportunity for reflection!

Reviewer 3 Report

The current study explored the use of hypothetical case scenarios related to teaching about the seafood industry to determine the use of these tools as a mechanism for increasing undergraduate students’ systems thinking capacity. However, there are serious concerns related to the relevance of paper with the scope of journal.

1. The paper describes the experience of students regarding understanding of systems complexity. 

2. Implications for using hypothetical case scenarios in the agricultural and natural resource education classroom are also explored.

How does the topic relevant with the scope of sustainability. The paper will be more relevant in the field of education or curriculum development.

References are also not updated.

The tools used are good enough for better understanding of students but it does not contribute to sustainable development.

Author Response

How does the topic relevant with the scope of sustainability. The paper will be more relevant in the field of education or curriculum development.

  • ACTION: We have attempted to add verbiage related to the SDGs to align the educational intervention with SDGs, outlining the specific SDGs relevant to the HCS curriculum. We have also outlined its relevance to blue food systems.

References are also not updated.

  • ACTION: We have provided two copies of the revised manuscript. One has the tracked changes for the review, the second is reformatted according to journal requirements.

The tools used are good enough for better understanding of students but it does not contribute to sustainable development.

  • ACTION: We have attempted to add verbiage related to the SDGs to align the educational intervention with SDGs.

Reviewer 4 Report

Two minor revisions

what are he emerging identities of Gen Z (on p. 6)? say more or delete

about how many experts vetted the questionnaire? p.14

Author Response

Thank you for your helpful review. Please see our responses to your two suggested edits below:

We have clarified these "emerging identities" as “sustainability identities” and added a citation supporting this assertion.

Three experts reviewed the focus group questionnaire and five reviewed the survey instrument. We have added this in the methods section.

Round 2

Reviewer 1 Report

This is the second review and I am afraid the paper shows the same shortcomings, some specific improvement excepted. Specifically, you have made some changes, such as including the sampling procedure, describing the measuring instruments, and removing some of the theoretical contents from the methodology section. Similarly, you have put forward new lines of research in the conclusion section.

However, not only do the most severe drawbacks remain, but it also has new problems as follows:

(1)    The introduction section lacks research objectives.

(2)    The introduction section does not append an advance of the paper structure. Besides, I suggest you adopt a classic design divided into introduction, literature review, methodology, results analysis, discussion and conclusion sections.

(3)    Consequently, the review of the literature is devoid of direction. It shows new subheadings but is not in line with any research objectives.

(4)    There is no clear distinction between the methodological and the analysis of the result sections. While the former should be dedicated to explaining the research techniques' context and applications, the measuring instruments, the target research population and elements, and the sampling procedure and units, the latter should display the obtained empirical evidence. Consequently, nearly 50% of the paper is plagued with methodological ingredients. Please, separate these different sections.

(5)    Not only do you raise the research questions out of the introduction section, but it is also devoid of any previous justification.

(6)    The paper presentation is neither neat nor tidy because there are many marks and crosses out. Moreover, its size is too long and more than 40 pages. Therefore, it is tiring to read. Please, be more concise, straightforward and efficient. Let me suggest you create a version free of crossing out and marks.

I hope these comments help improve the paper and encourage you to progress.

Author Response

The introduction section lacks research objectives.

Due to the use of mixed method design, we have elected to use research questions. The purpose and research questions are now at the end of the introduction.

The introduction section does not append an advance of the paper structure. Besides, I suggest you adopt a classic design divided into introduction, literature review, methodology, results analysis, discussion and conclusion sections.

We appreciate the reviewer’s feedback. However, we have followed the “Research Manuscript Sections” guidance as outlined in the “Instructions for Authors” on the journal website.

Consequently, the review of the literature is devoid of direction. It shows new subheadings but is not in line with any research objectives.

We believe our introduction relates to research question 2 (systems thinking perspective) and our theoretical framework relates to research question 3 (transformative learning theory). However, we have adjusted the titles of the subheadings to better connect them to the RQs.

There is no clear distinction between the methodological and the analysis of the result sections. While the former should be dedicated to explaining the research techniques' context and applications, the measuring instruments, the target research population and elements, and the sampling procedure and units, the latter should display the obtained empirical evidence. Consequently, nearly 50% of the paper is plagued with methodological ingredients. Please, separate these different sections.

We are unclear exactly which components of the methods the reviewer would like us to remove from the results section. The first section of the results contains frequencies and the subsequent sections contain qualitative results, aligned with manuscript norms for social science/qualitative writing. However, we are happy to make adjustments if you could help us identify the 50% of the manuscript you are referring to. We have attempted to remove verbiage related to methods in the results section.

Not only do you raise the research questions out of the introduction section, but it is also devoid of any previous justification.

We have integrated the research questions into the introduction section.

The paper presentation is neither neat nor tidy because there are many marks and crosses out. Moreover, its size is too long and more than 40 pages. Therefore, it is tiring to read. Please, be more concise, straightforward and efficient. Let me suggest you create a version free of crossing out and marks.

We apologize for the difficult nature of reading the manuscript; however, we complied with journal requirements to conduct all revisions in tracked changes in Microsoft Word.

Reviewer 3 Report

This topic is beyond the scope of journal hence rejected

Author Response

Thank you for your review. However, other publications in the journal have similar themes and we hope justify our position within the scope (see Gibson et al., 2023, https://doi.org/10.3390/foods12101933) as well as other articles focusing on consumer behavior related to sustainable food systems (see Ma & Chang, 2022, https://doi.org/10.3390/foods11162423)